# Transient Effect at the Onset of Human Running

**DOI:** 10.3390/bios10090117

**Published:** 2020-09-08

**Authors:** Christian Weich, Manfred M. Vieten, Randall L. Jensen

**Affiliations:** 1Sports Science, University of Konstanz, 78464 Konstanz, Germany; manfred.vieten@uni-konstanz.de; 2School of Health & Human Performance, Northern Michigan University, Marquette, MI 49855, USA; rajensen@nmu.edu

**Keywords:** attractor method, kinematics of human cyclic motion, motion analysis, transient effect, accelerometer

## Abstract

While training and competing as a runner, athletes often sense an unsteady feeling during the first meters on the road. This sensation, termed as transient effect, disappears after a short period as the runners approach their individual running rhythm. The foundation of this work focuses on the detection and quantification of this phenomenon. Thirty athletes ran two sessions over 60 min on a treadmill at moderate speed. Three-dimensional acceleration data were collected using two MEMS sensors attached to the lower limbs. By using the attractor method and Fourier transforms, the transient effect was isolated from noise and further components of human cyclic motion. A substantial transient effect was detected in 81% of all measured runs. On average, the transient effect lasted 5.25 min with a range of less than one minute to a maximum of 31 min. A link to performance data such as running level, experience and weekly training hours could not be found. The presented work provides the methodological basis to detect and quantify the transient effect at moderate running speeds. The acquisition of further physical or metabolic performance data could provide more detailed information about the impact of the transient effect on athletic performance.

## 1. Introduction

In general, human motion, even the movements which are repeated many thousand times, e.g., by athletes, cannot be called absolutely consistent and stable [1,2]. No single movement is like any other and they are always characterized by a high degree of individuality [3,4]. Beyond the actual movement, it is also apparent that changes in movement patterns have an influence on the subsequent motion kinematics and thus must be highly controlled and regulated by the body and the brain, respectively. Most obvious is the change from a resting situation like sitting to physical activity [5,6] or the change between two forms of movement, like from walking to running and back [7,8] or cycling to running in triathlon [9,10,11]. Weich et al. [5] showed in a triathlon study, not only that the transition run after cycling showed deviating behavior over the first minutes of the session, but also from the control condition, an isolated 5000 m run. Either way, athletes often sense an irregular or uneven way of running at the onset of their exercise or in these post cycling performances in triathlons and report that this phenomenon commonly subsides within a few minutes [9,12]. This phase of finding-a-rhythm might be related to the well-known transient oscillations described in dynamical systems [13]. Respecting scale invariance, numerous examples, such as analytical equations [14], human neurology [15], as well as biomechanics [16], are characterized by this behavior. If any dynamical system is initiated or affected by an internal or external perturbation, it takes some time to even out to a balanced condition. The asymptotical equilibrium reached after these perturbations can be called an attractor [17,18]. The crossed trajectories, i.e., the paths of the systems’ states over time, are called transients. Once initialized, these transients show rapidly changing and mostly irregular behavior over a short-lived period until settling down to the attractor [18]. Based on these observations, human cyclic motions, such as walking or running, can be described as limit-cycle attractors [19,20,21]. An approach to analyze attractors, the attractor method, derived from human cyclic motion was introduced by Vieten et al. [21]. The latter yields very sensitive results allowing the analysis of subtle changes of movement patterns and their variations. The application and further studies assessing athletes while undisturbed running and cycling indicated the existence of a transient effect at the onset of physical activities [9]. Recently Vieten and Weich [12] extended these earlier findings by demonstrating a mathematical model of the kinematics of human cyclic motion when considering transient oscillations a crucial component of locomotion. The starting value of the deflection is very subject-specific and influenced by randomness. Nevertheless, progress from the beginning of the run until finally levelling off can be modelled as the solution of a damped harmonic oscillator decreasing with a negative exponent as a function of time. Once a runner has reached this point, the transient effect remains, but its extent is reduced to a level that is subjectively no longer perceptible to the athlete [12].

Based on the mathematical model, the aims of the present study were to determine the existence of the transient effect and to quantify it in athletes running at moderate speed. The analysis focused on the magnitude and duration of the movement’s transient fluctuations and their subject-specific characteristics. It was also considered whether training level, athletic experience, or anthropometric preconditions were related to this phenomenon. Insights emerging from this research may provide new aspects concerning the nature of running, opening new possibilities for race pacing and overall performance.

## 2. Materials and Methods

A total of 30 athletes (Table 1), 10 female and 20 males, were tested from October 2019 until June 2020 in Kreuzlingen, Switzerland (Elitesportschule Thurgau). All participants were regularly physically active, and none were suffering any current injury, which would have impeded their performance. The only prerequisites were to be able to run for 60 min without reducing their expected performance which had been determined in advance. Furthermore, training level and experience, using the training hours per week and the number of years the athletes had been running, were obtained. The study was approved by the local Ethical Committee of the University of Konstanz, Germany, under the Ref. No: IRB20KN08-001. All participants were requested to fill out and sign an informed consent.

To collect the necessary raw accelerometer data, two inertial sensors were used (RehaWatch by Hasomed, Magdeburg, Germany). The sensors have a size of 60 × 35 × 15 mm and weigh 35 g each. They function as a triaxial accelerometer with up to 16 G, a triaxial gyroscope with up to 2000°/s and a magnetometer measuring with 1.3 Gauss. The device is constructed as a micro-electro-mechanical-system (MEMS). For the current study, the MEMS measured the acceleration of the feet in three dimensions (x, y, z) with data saved to a smartphone (J5 by Samsung, Seoul, Korea). To collect the running motion, the sensors were attached to both ankles above each lateral malleolus by a hook-and-loop fastener. The runs were performed on a treadmill (9500HR by Life Fitness, Unterschleißheim, Germany).

Each runner had to run two sessions at a constant speed over 60 min. Before they started the first test session, they were asked to self-select their running pace, defined by a subjective feeling associated with a BORG CR-10 scale value of 3 to 4, i.e., moderate to somewhat severe [22]. Afterwards they were equipped with the two activated MEMS sensors attached to the ankles as described above. The treadmill was set at 1% inclination, to simulate air resistance, and the individual running speed. Once the speed was reached the data collection was started right before the runner jumped smoothly from the lateral standing area of the treadmill to the actual moving belt (the jump was subsequently removed by cutting out the first 1.5 s of the data set). During the sessions the smartphone was placed beside the treadmill. The measurements were received at a sampling frequency of 500 Hz assembled over a period of 60 min without interruption.

Recently, Vieten and Weich [12] introduced a model to mathematically describe the kinematics of human cyclic motion. Based on their model, six elements contribute to overall human cyclic motion. Here, the subject’s individual attractor $\vec{A}\left(t\right)$, a limit cycle in acceleration space, which is repeated in each cycle, is by far the biggest contributor. In addition, short-term fluctuations (random walk) around the morphed attractor, a control mechanism that regulates the latter if the current movement accelerations deviate too much from the attractor and technical (white) noise from the MEMS sensors contribute to the overall movement. These three components have an average contribution of zero due to cancellation processes and thus can be neglected in the context of this publication. Furthermore, the kinematics are markedly more affected by so-called attractor morphing $\vec{M}\left(t\right)$, a process which slowly changes the actual attractor and transient oscillations $\vec{T}\left(t\right)$. To reveal the morphing process $\vec{M}\left(t\right)$ and the transient oscillations $\vec{T}\left(t\right)$, the attractor must be subtracted from the measured signal $\vec{K}\left(t\right)$.
(1)$$\vec{K}\left(t\right)-\vec{A}\left(t\right)=\vec{M}\left(t\right)+\vec{T}\left(t\right)$$

These two terms, morphing and transient effect, are retained, when the reduced attractor is developed. So, the transient effect’s contribution can be written as
(2)$$\vec{T}\left(t\right)=\vec{K}\left(t\right)-\vec{A}\left(t\right)-\vec{M}\left(t\right)$$

Morphing $\vec{M}\left(t\right)$ contributes minor changes to the attractor over the time of the activity. Thus, the effect is generally bigger between different runs as compared to changes within a run. Because the transient effect decreases asymptotically within the first minutes, a super attractor $\vec{S}$ is created as the mean of all one-minute attractors of a single running session. This results in the closest approximation to the second and third terms of the right-hand side of Equation (2)
(3)$$\vec{S}=\langle \vec{A}\left(t\right)-\vec{M}\left(t\right)\rangle$$
and following the transient effect can be approximated by
(4)$$\vec{T}\left(t\right)=\vec{K}\left(t\right)-\vec{S}$$
which allows the calculation of the transient time from
(5)$$\frac{T\left(t_{trans}\right)}{T\left(t=0\right)}=e^{-1}$$

Since further analysis was required, the collected 60-min data block was divided into 60 s intervals using a file splitter to produce 60 single datasets. Afterwards the MATLAB app Attractor was used to calculate the attractor data of each one-minute data set. The functionality of the Attractor app is based on the Attractor Method developed by Vieten and colleagues [21] and is available online via http://www.uni-konstanz.de/FuF/SportWiss/vieten/CyclicMove/. Further, a super attractor $\vec{S}$ of each run was calculated to represent each participant’s individual gait-print as a mean attractor of all single minutes of each session [3]. To detect, quantify, and validate the transient effect of a running performance, three mathematical procedures were considered (Figure 1):

### 2.1. Delta M (δM, Method I)

The parameter δM represents the velocity (v) normalized average distance between two attractors and can be described as follows [12]
(6)$$\delta M=\frac{1}{v}\cdot \langle T_{∥}\cdot \left[e^{\frac{-t}{t_{T}}}-e^{\frac{-t_{E}}{t_{T}}}\right]+\;a_{0}\cdot \{\sqrt{\frac{\left(t_{E}-t\right)}{t_{E}}}+a_{1}\cdot sin\left(a_{2}\cdot 2\pi \frac{\left(t_{E}-t\right)}{t_{E}}\right)\}\rangle$$
where the given constants $T_{∥},t_{T},a_{0},a_{1},a_{2}$ are derived from a curve fitting application of all measurements (CurveExpert Professional 2.6.5 (version 2.6.5, Hyams Development), using the Levenberg−Marquardt algorithm.

The constants $a_{0},a_{1},a_{2}$ represent the morphing process whereas $T_{∥}\;\mathrm{and}\;t_{T}\;$ are based on the transient oscillations at the onset of a movement [12]. $t_{T}$ quantifies the time until the oscillation decreases to $e^{-1}$ of its original starting value T. The super attractor $\vec{S}$ of each running session was taken as a stable condition to be compared to each single minute from the start of the exercise bout until the end of minute 60. Since the super attractor, by definition, represents only one cycle and the Attractor app can only compute multiple cycles in one data set, each one was extended to a data set with a total duration of one minute. For this data set, which is designated as a super minute, the original cycle is repeated according to the average cycle length. The starting value of the oscillation is highly dependent on the individual and is further affected by random processes. The average distance between two attractors, defined as δM, for the same subject, is small, compared to attractor differences between different subjects [3].

Accordingly, the result of this analysis provides a δM value for each comparison between one of the single minutes and the super minute, so that a list of sixty δM values in temporal sequence describes the transient process over the entire session (Figure 1, red curve). A smaller δM number means a higher similarity of the compared attractor pairs. The latter procedure is executed using the Attractor app comparing the super attractor $\vec{S}$ (in the super minute version) with the measured acceleration data leading to the transient time T(t) when further processed with the curve fitting software (CurveExpert Professional 2.6.5 (version 2.6.5, Hyams Development)).

### 2.2. Fast Fourier Analysis (FFT, Method II)

The Fast Fourier Analysis (FFT) [23] was used as a second possibility to quantify the transient effect (Figure 1, blue curve). This enables the recorded data to be viewed in the frequency domain to allow the possibility of filtering to maintain only the essential data choosing a suitable cutoff of 10 Hz. In both, frequency and time space, the L^2^ norm is used. Based on the Plancherel theorem [24], which states that for L^2^ functions the norm of the time domain is retained in the FFT (frequency domain), the transient time can be described as below:(7)$$\overset{+\infty }{\underset{-\infty }{\int }}{\left|g\left(t\right)\right|}^{2}dt=\overset{+\infty }{\underset{-\infty }{\int }}{\left|F\left(f\right)\right|}^{2}df$$

This allows the calculation of the transient effect using the Fourier transform of the reduced signal as a measure of the discrepancy of the measured running minutes $\vec{K}\left(t\right)$ from the super attractor $\vec{S}$, which represents the athlete’s average running behavior. The calculation of a scalar is independent of the used coordinate system. It follows:(8)$$T\left(t\right)=\sqrt{\overset{t_{e}}{\underset{0}{\int }}{\left(\vec{K}\left(t\right)-\vec{S}\right)}^{2}dt}=\sqrt{\overset{f_{c}}{\underset{-f_{c}}{\int }}F^{2}\left(f\right)df}$$
where $f_{c}$ is the cut off frequency (here 10 Hz), because the information content of a filtered signal is considered and $t_{e}$ representing the final minute of a running session. The transient effect is treated as a damped harmonic oscillator [12]. As a temporary oscillation around the attractor, these oscillations have contributions corresponding to the harmonics up to the cut off frequency only. All other contributions mentioned above, those of the frequencies different from the harmonics, naturally cancel out due to destructive interference. Thus, the numerical calculation of the right-hand side of Equation (7), is executed as adding up all amplitudes of the harmonics up to the cut off frequency which was set at 10 Hz.
(9)$$T\left(t\right)\sim \sqrt{\overset{n\left(f_{c}\right)}{\underset{i=1}{\sum }}F^{2}\left(f_{i}\right)}=CoH\left(t\right)$$

The expression (CoH(t): collection of harmonic amplitudes) is proportional to the transient contribution and is impacted very little by other contributions to movement. The attractor and morphing, which change rather slowly (≪1 Hz), were subtracted before the Fourier transform. Furthermore, the residuals do not contribute very much because their contribution is almost always different from the harmonics. The contributions of the other movement parts (noise, short-term fluctuations and the control mechanism) have contributions that differ from the harmonics and thus are not subtracted. As a consequence, the Fourier transformed expression allows for a more accurate calculation of the transient time, $t_{trans}$, using curve fitting software (CurveExpertPro, version 2.6.5, Hyams Development).
(10)$$\frac{CoH\left(t_{trans}\right)}{CoH\left(t=0\right)}=e^{-1}$$

### 2.3. Modified Fast Fourier Analysis (FFTmod, Method III)

Another, third, way of calculating T(t) can be carried out as follows:(11)$$\begin{array}{l} T\left(t\right)=\sqrt{\overset{t_{e}}{\underset{0}{\int }}{\left(\vec{K}\left(t\right)-\vec{S}\right)}^{2}\;dt}=\sqrt{\overset{t_{e}}{\underset{0}{\int }}\left(K^{2}-2\;\vec{K}\cdot \vec{S}+S^{2}\right)\;dt}\\ \;=\sqrt{\overset{f_{c}}{\underset{-f_{c}}{\int }}F_{K^{2}}\left(f\right)\;df-2\overset{f_{c}}{\underset{-f_{c}}{\int }}F_{KS}\left(f\right)\;df+\overset{f_{c}}{\underset{-f_{c}}{\int }}F_{S^{2}}\left(f\right)\;df} \end{array}$$
where the FFT is executed first for all components, before the results were combined (Figure 1, green curve). This calculation differs from the FFT method (Equation (8)) such that by applying the Fourier transform before the subtraction, the other unwanted components of cyclic human motion in $\vec{K}\left(t\right)$ and $\vec{S}$ [12] are separated from the actual transient effect. This is valid because the contributions of the transient effect are only present in the harmonics, which remain after the Fourier transform.
(12)$$T\left(t\right)\;\sim \sqrt{\overset{n\left(f_{c}\right)}{\underset{i=1}{\sum }}F_{K^{2}}\left(f_{i}\right)-2\overset{n\left(f_{c}\right)}{\underset{i=1}{\sum }}F_{KS}\left(f_{i}\right)+\;\overset{n\left(f_{c}\right)}{\underset{i=1}{\sum }}F_{S^{2}}\left(f_{i}\right)\;}=CoHmod\left(t\right)$$
where CoHmod(t) (collection of harmonic amplitudes from the FFTmod method) is proportional to the transient contribution and is not impacted by the other contributions of the movement. Again, the Fourier transformed expression allows for a more accurate calculation of the transient time, $t_{trans}$, using curve fitting software. Therefore equation 10 can be used applying harmonic amplitudes resulting from the FFTmod calculations.

It was still expected that the transient times of both FFT methods would be very close together. For this reason, a correlation was also calculated for this relationship.

When calculating the transient time T(t) using method I, the transient effect and the residuals of the described components of human cyclic motion (morphing, short-time fluctuations, control mechanism, noise) [12] are included. The Fourier transform, on the other hand, allows the two other applications (II and III) to isolate the transient effect, because the latter is found mainly in the harmonics. Thus, an intraclass correlation (executed with SPSS version 26.0) was used to calculate the strength between method I and II/III. This statistical method provides a measure of the proportion of variance that is attributable to the objects being measured. Consequently, to be able to make a decision about the presence of the transient effect, method one (I) and at least one of the two FFT methods (II or III) must be computed. A high ICC implies that most of the variance is among group/method. If the test reveals a high intraclass coefficient (ICC), r means that the checked running data show a substantial transient effect (Figure 2, black data). A low ICC (Figure 2, red data) would rather reflect the absence of initial oscillations. In order to be able to determine the strength of the resulting ICC, the categorization according to Hopkins [25] was used. Hopkins expands on the original work of Cohen (1988) by further classifying coefficients greater than 0.5 (strong) into very high (r = 0.7−0.9) and almost perfect or indistinguishable (r > 0.9). For the present work, an r > 0.7 is regarded as a suitable magnitude to assume the existence of a substantial transient effect. Furthermore, this corresponds to a coefficient of determination (R^2^) above 0.49, which explains at least 49% of the variance.

In a final step the collected anthropometric and performance data were statistically analyzed via correlation (executed with SPSS version 26.0) to determine whether running experience is related to the observed transient times.

## 3. Results

Of the 60 sessions (two for each subject), 48 (for FFT) and 49 (for the FFTmod) cases showed a transient behavior at the onset of the running session (Figure 3). This corresponds to 80 and 81% of the cases considered, respectively. It can be observed that for FFT in two and for FFTmod in one case, both runs of a person showed no transient effect (subject 27 for both methods, and also 13 for FFT). Another eight participants (2, 3, 6, 9, 16, 21, 25, 28) had a mixed result (for FFT and FFTmod) and showed one run with and one without a transient effect. Only one person (13) offered a mixed result for FFTmod but displayed no transient effect for both runs using the FFT. In general, the correlation (ICC) between the FFT methods was calculated with a mean r = 0.99, indicating a high accordance of both approaches.

Furthermore, the extent or duration of the transient effect was calculated for those runs that displayed a transient effect (48 cases for FFT and 49 for FFTmod). On average, the initial transient oscillations, i.e., the time it takes athletes to find their rhythm, took 4.99 (±3.35) minutes when the data were evaluated with the FFT method, and slightly longer, 5.50 (±4.72) minutes using the FFTmod method (Figure 4). The transient effect ranged from 31 min (subject 8) on the higher end to less than one minute (subject 17, run 1; subject 20, run 2) for the shortest.

To determine whether anthropometric or performance characteristics were related to the transient time, only runs that showed a transient effect (according to the above explained analysis methods) were considered. As Table 2 shows, none of the recorded anthropometric or performance measures were significantly correlated to transient time (*p* > 0.05).

## 4. Discussion

The main objective of the present study was the detection and quantification of the transient effect in human running. It was shown, that depending on the analysis method, this effect occurred in 80−81% of the running sessions in the participating subjects. The average time until an athlete found a running rhythm was 4.99 (FFT) to 5.50 (FFTmod) minutes, with the longest adaptation time being 31 min. Furthermore, the results of the FFTmod (Equation (12)) can be classified as clearer, since the Fourier Transform separates the morphing and other components of cyclic human motion from the transient effect before combining the parts of the mathematical equation. Both the occurrence and the duration of the transient effect were independent of body characteristics and performance measures such as running level, weekly training hours, and training age (all *p* > 0.05).

The high appearance of the transient effect in more than 80% of the recorded runs suggests that the transient effect is a quite common phenomenon within the context of the participating group, the performance level, and a moderate running pace. The current participants may be expected to be representative of the general running community, as they represent a high range of age (18 year–55 year), training experience (1 year–40 year), running performance expressed as the pace for a moderate endurance run (8 km/h–15 km/h) and weekly training hours (0.75 h–10.5 h). This is further supported by the fact that none of the mentioned specific performance or body data are significantly related to the occurrence of the transient effect.

The duration of the transient effect is defined as the time required to reduce the magnitude of the initial state of the transient value (Figure 1, first data point of the curves) to e^−1^. The average time derived from the data of this study was 5.25 min, which is consistent with reported observations of experienced runners. In addition to these observations, there is also a scientific data base that describes a similar time frame for changes in running rhythm in related contexts like the transitioning from cycling to running in triathlon [26,27] or from walking to running and back [7,8]. As an example, Gohlitz et al. [26] reported differences in adaption time of the prior working muscular conditions in cycling and running. Their main finding was that immediately after the transition the athlete’s body takes a while to adapt to the varying motion. During this phase, lasting around 1200 m, the stride frequency decreased, and the stride length increased until the participants regained their personal optimum (measured earlier in another 5000 m run). This adjustment was also consistent with their subjective feeling during the run. Witt [27] confirmed these results and tried to interpret them with physiological and biomechanical explanations after testing triathletes during a run-cycle-run condition. The author claimed that cycling destroyed the activity pattern of a subsequent run due to extremely different working conditions of the muscles between both disciplines. Later, Weich et al. [9] published a cross-over study with triathletes based on this idea and compared isolated running over 5 km with a run of the same length after prior cycling. In this context, they noted that there was not only, as expected, a transition phase when running after preload, but also during the solo run. The average duration mentioned in this paper was 7 min for both types of running. Here, too, the authors proposed predominantly neuromuscular reasons for the initial transient effect. It should certainly remain a major objective in future studies to show the exact origin of these phenomenon.

Even though the vast majority of the runs in the current study showed a transient effect, there are also participants who either had no initial oscillations or a mixed outcome with one run showing an effect whereas the other had none. This has also been shown in previous studies [3,12]. Vieten and Weich reported, that the starting value of the deflection is very subject-specific and influenced by randomness. It can therefore be assumed that, especially in the case of a mixed result for one person, the athlete found his or her rhythm by chance and was close to his or her individual attractor right from the beginning. Thus, the athlete did not undergo a prominent transient effect, which, as a consequence, was not visible. Further a subject-inherent property in the motor control system could be responsible. Systematic (or nonsystematic) fluctuations over the entire run have the same magnitude as the transient effect, so that it does not appear prominently (see Figure 5 for subject 9, session 1). In this case, it could be because the subject has, with only 1.5 years and 2.5 weekly training hours relatively little running experience. In simple terms, this athlete repeatedly experiences deviations from his/her running rhythm, which he/she has to gain back over and over again. One could almost say that these fluctuations are multiple small transient effects, which the subject has to overcome. The cause for these fluctuations may also lie in other components of cyclic human motion, e.g., morphing as well as the residuals of short-time fluctuations, control mechanism and noise. They can be so pronounced that they obscure the transient process. A further scenario can be seen in athlete 13, who is almost a professional runner experiencing no transient effect in any of the runs. Here the assumption is quite reasonable that he/she is able to find the running rhythm very rapidly due to the high level of performance and many thousands of kilometers of yearly running (more than 100 km per week). However, to confirm this assumption, a separate study with multiple professional runners at the same or even better level is needed. In general, if the initial numerical value of a session is already very low (as in Figure 5, only 1.5), this means that the data is very similar to the general trend (represented by the super attractor). Thus, it is very likely that the transient effect will not be visible.

On the other hand, it sometimes happened that there were slight disturbances (e.g., a short stumble) during a session, but these were smaller than the transient effect and so short that the running behavior was not affected. If the athlete is impaired by minor, short-term disturbances (Figure 6, for example minute 17, marked with the black arrow), the presented analysis method seems to be robust to the extent that the system balances out again within a short time period.

In a practical sense, the question arises as to what influence the transient effect at the beginning of a training session or competition has on running performance. To be able to clarify this question with certainty, it is necessary to examine metabolic or physiological data, such as oxygen uptake, heart rate, electromyography etc., which would provide further insights into the course of events happening during the initial phase of the exercise. From this study, anecdotal evidence suggests that running (without warm-up) was perceived as more comfortable after a few minutes into the run. If the transient effect should have a negative impact on running performance, this could be another powerful argument for an extended warm-up before each training session or running competition [28,29]. This would allow the athlete to be in his or her individual running rhythm at the start of the race or the main part of the training.

In order to strengthen the general validity of this phenomenon, future studies should validate the outcome of the test procedure in an outdoor setting and at varying or self-selected running speeds. Furthermore, the analysis in this study was carried out only with experienced nonprofessional runners. It remains open how the transient effect behaves in novice or professional runners. It could also be interesting to include other cyclic sports such as swimming, cycling, walking or rowing.

## 5. Conclusions

In summary, it can be concluded that by applying the attractor method and the described analysis process of the data in frequency space, a transient effect can be detected in over 80% of the recorded running sessions. On average, the initial oscillations lasted 5.25 min, which correspond to about 500−1000 m, depending on running ability. This also corresponds to the subjective feeling that athletes report empirically. For the runners who did not experience a transient effect, other components of their running kinematics, such as morphing, might have hidden the phenomenon. Because the FFTmod method contains predominantly fractions of the transient effect, it produces the most precise outcome and is therefore the recommended method. Taking into consideration further physiological and metabolic data in future works will offer the chance to determine the influence of this phenomenon on athletic performance in training and competition.

## Figures and Tables

**Figure 1 biosensors-10-00117-f001:**
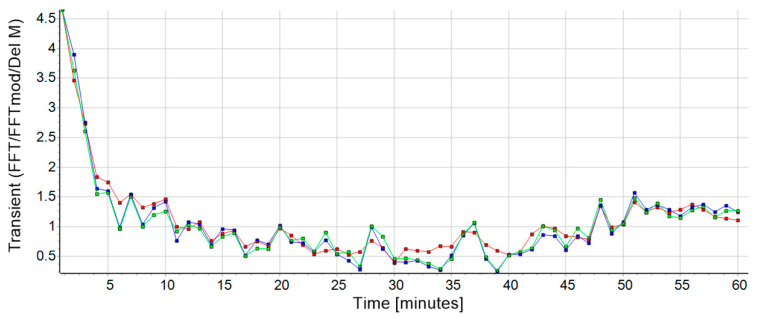
Outcome (subject 26, session 1) for the transient values from all three calculation methods: δ*M* in red, FFT in blue and FFTmod in green. The transient effect can clearly be seen at the onset of the run. The data from FFT and FFTmod were normalized to the data from δ*M* to improve the visualization of the comparison. This had no influence on the result.

**Figure 2 biosensors-10-00117-f002:**
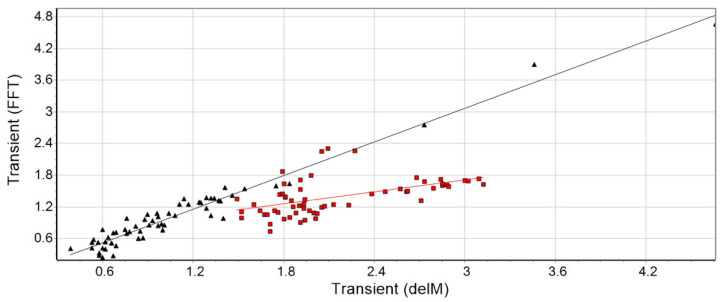
Examples of two different subjects, one with a high correlation (subject 29, session 1, r = 0.99, black triangles) showing a transient effect and one with a low correlation (subject 21, session 2, r = 0.60, red squares) without a distinct transient phase at the beginning.

**Figure 3 biosensors-10-00117-f003:**
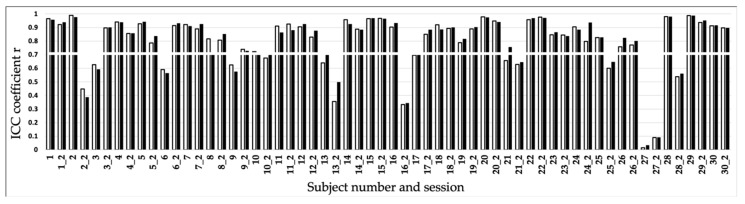
Overview of the transient effect for all subjects across the two sessions listed for example as: 1 = subject 1 session 1; while 1_2 = subject 1 session 2. White bars with a frame represent the ICC of the FFT method and the filled black bars the FFTmod method. The white horizontal line displays the critical r of 0.7. Correlations reaching above the white horizontal bar were considered to have a detectable transient effect.

**Figure 4 biosensors-10-00117-f004:**
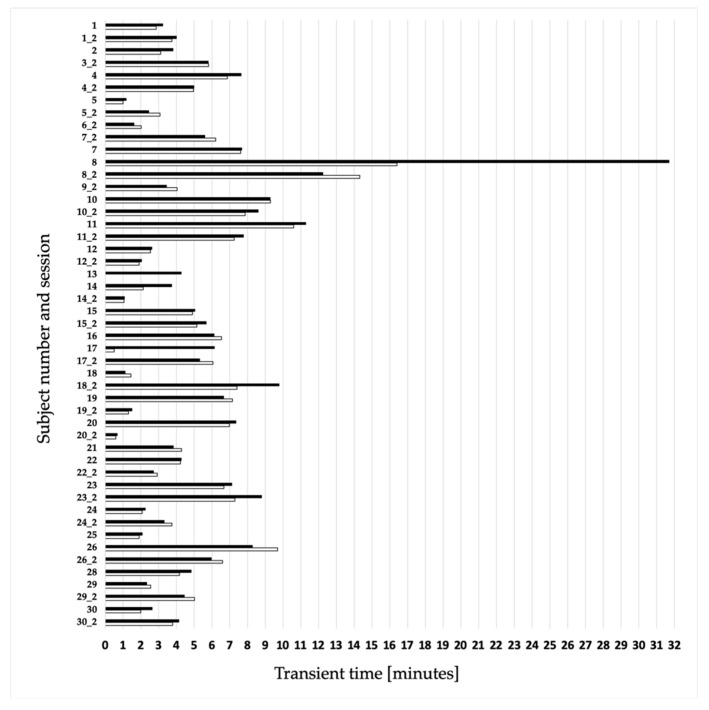
Overview of the transient effect for all subjects across the two sessions listed, for example, as: 1 = subject 1 session 1; while 1_2 = subject 1 session 2. White bars with a frame represent the transient time in minutes calculated by the FFT method and the filled black bars the time derived by the FFTmod method. All cases without detectable transient oscillations at the onset of their run were excluded.

**Figure 5 biosensors-10-00117-f005:**
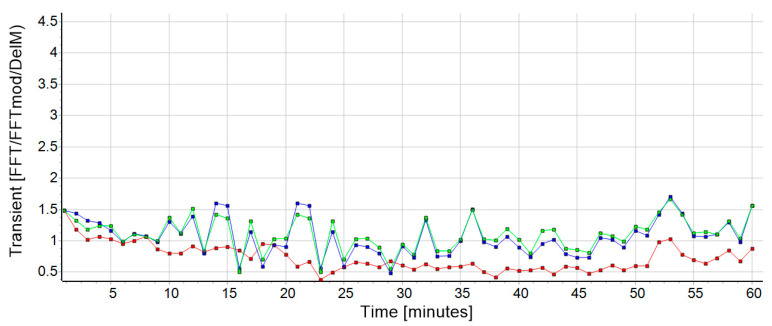
Outcome (subject 9, session 1) for the transient values calculated by δ*M* in red, FFT in blue and FFTmod in green. In this example, there is no transient effect, since it is probably overlaid by strong fluctuations of the same magnitude over the entire run (r = 0.6). The y-axis scale was adapted to Figure 1 to highlight the difference between the starting values. The low numerical transient value of 1.5 here indicates that most probably no transient effect will be detected. The data from FFT and FFTmod was normalized to the data from δ*M* to clarify the visualization of the comparison. This had no influence on the result.

**Figure 6 biosensors-10-00117-f006:**
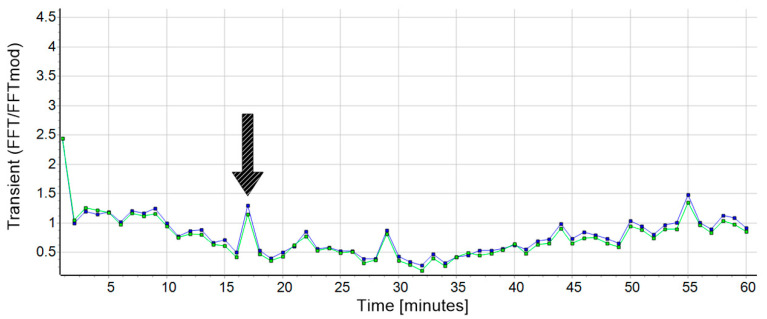
Outcome (subject 22, session 1) for the transient values calculated by FFT in blue and FFTmod in green. In this example, the data show a transient effect (transient time = 4.23 min (FFT) and 4.3 (FFTmod); r > 0.9), although the course of the session shows minor disturbances (like minute 17 (black arrow), which were controlled within a short time period. The y-axis scale was adapted from Figure 1 to highlight the difference between the starting values. The data from FFT and FFTmod was normalized to the data from δ*M* (not shown) to improve the visualization of the comparison but had no influence on the result.

**Table 1 biosensors-10-00117-t001:** Subject overview.

	N	Age (y)	Height (cm)	Weight (kg)	Test Speed (km/h)	Training (h/Week)	Running Experience (year)
**Female**	10	29 ± 11.5	166 ± 5.5	55.7 ± 4.0	9.6 ± 1.4	3.6 ± 1.8	10.9 ± 8.3
**Male**	20	29 ± 11.1	180 ± 5.6	70.1 ± 6.7	11.5 ± 1.5	4.6 ± 2.2	10.7 ± 10.9
**Overall**	30	29 ± 11.3	175 ± 8.9	64.9 ± 9.1	10.8 ± 1.4	4.2 ± 2.1	10.8 ± 9.8

**Table 2 biosensors-10-00117-t002:** Correlations transient time and performance/anthropometric data.

			r	*p*
Transient time [min]	-	Age [year]	−0.010	0.958
Transient time [min]	-	Weight [kg]	−0.303	0.104
Transient time [min]	-	Height [m]	−0.015	0.937
Transient time [min]	-	Running experience [year]	0.086	0.653
Transient time [min]	-	Weekly training [h]	−0.045	0.815
Transient time [min]	-	Test running velocity [km/h]	−0.093	0.626

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
