# Peer review of "Transient Effect at the Onset of Human Running"

_biosensors, 2020, doi:10.3390/bios10090117_

Round 1

Reviewer 1 Report

There is considerable interest to determine the influence of this phenomenon on athletic performance in training and  competition. This manuscript describes the detection and quantification of transient effect - an unsteady feeling of athletes during the first meters on the road. The authors used MEMS sensors to collect acceleration data and employed attractor approach and Fourier transforms to  determine the transient effect which was in 81% of the runs. Some useful information was drawn from this work, including the time of transient effect in athletes.

Overall this is a well written manuscript. The reviewer think research methods were appropriate and the analysis was well conducted. I think the authors could give more depth discussion on the two different subjects with and without transient effects.

Figures could be revised to make them readable, e.g. font size.

Author Response

Thank you very much. We highly appreciate your review and comments.

Reviewer 2 Report

The authors describe a method in order to identify an initial transient effect on a runner and to measure its duration. The paper is interesting, there is a huge amount of results, that are analized according to different methods.

There are some relevant aspects to be clarified:

1) according to the contents of the manuscript it is not clear to me the final aim of the paper; either validation of the proposed method of the attractor method, previously proposed by the authors, by demonstrating the capability of identifying the transient effect or the identification of  transient effect itself. it is not convincing to say that the transient effect does'n occur in cases when the method does'n identify it.

2) the agreement of different three approaches seems very satisfactory in cases the transient effect is found: which is the estimated difference among methods? could it depend on the effect of M(t)?

3) in the paper (section "Results") session 1 and session 2 are mentioned: no mention about this in session 2 "Materials and methods"; how many tests wre carried on for each runner?

4) are differences between the indications of each MEMS with reference to the same runner?

Minor remarks

1) typos in rows 39-4 (Introduction".

2) in formula (6) some quantities are not defined in the text.

Author Response

(The authors gave the same response as above.)

Round 2

Reviewer 2 Report

No matter with respect to minor remarks, now formula (6) is more clear.

About answers to my comments concerning major questions:

the Authors confirm concepts, which are already clear in the paper, I agree. The FFTmod method is more accurate, due to the fact it is unaffected by unwanted contributions.

My comment is that, in my feeling, the transient effect, T(t) can be acknowledged and measured, when unwanted not initial transients, M(t) part of movement and other disturbing effects are lower than a threshold, making approximations of method 1 negligible: therefore ICC is higher than 0.7 and transient acknowledged.

In order to making more clear the paper, the first sentence" The existence of a transient effect was assumed to occur when the ICC r > 0.7 (p < 0.05) was
determined between the three methodologies described above." of the Results section should be express clearly as a part of the methodology and moved consequently. Furthermore, it should emphasized in the Method's section that the methodology requires expressly the use of all computation methods, even though the FFTmod is the most accurate.

Finally, results about transient are interesting and useful for further improvement of treatment of athletes.

Author Response

Thank you very much for your comments.
